# NON-ITERATIVE PARALLEL TEXT GENERATION VIA GLANCING TRANSFORMER

## ABSTRACT

Although non-autoregressive models with one-iteration generation achieve remarkable inference speed-up, they still fall behind their autoregressive counterparts in prediction accuracy. The non-autoregressive models with the best accuracy currently rely on multiple decoding iterations, which largely sacrifice the inference speed of non-autoregressive models. Inspired by the way of learning word dependencies in autoregressive and iterative-decoding models, we propose Glancing Transformer (GLAT) with a glancing language model (GLM), which learns to capture the word dependency gradually. Experiments on three benchmarks demonstrate that our approach can significantly improve the accuracy of non-autoregressive models without multiple decoding iterations. In particular, GLAT achieves state-of-the-art results among non-iterative models and even outperforms top iterative counterparts in some specific benchmarks.

## 1 INTRODUCTION

Non-autoregressive transformer (NAT) has attracted wide attention in neural machine translation (Gu et al., 2018), which generates sentences simultaneously rather than sequentially. To enable parallel decoding, NAT imposes a *conditional independence assumption* among words in the output sentences, which leads to significantly faster inference speed (almost a dozen times speed-up) than the autoregressive Transformer (Vaswani et al., 2017). However, NAT still falls behind autoregressive Transformer (AT) in the quality of output sentences, such as BLEU (Papineni et al., 2002) for machine translation. We blame it for the imposed conditional independence assumption, which prevents NAT models from explicitly learning the *word dependencies* in the output sentence. Note that such word dependency is crucial, and it is explicitly learned in the AT model through the *autoregressive language models* (left-to-right, see Figure 1a and Figure 1b).

Recently, Ghazvininejad et al. (2019); Gu et al. (2019) propose to employ the *Masked Language Model* (MLM, Devlin et al., 2019) in NAT, which includes word dependency modeling in an *iterative* fashion (see Figure 1c), therefore yielding quite competitive results compared to AT. Specifically, such iterative models randomly mask words in the reference and predict these masked words conditioned on unmasked ones during training. In this manner, iterative models are trained to explicitly capture the dependencies between masked words and unmasked words.

However, these iterative approaches still give poor results with one decoding iteration and have to perform multiple iterations during inference, namely iteratively refining the generated outputs of the previous iteration. Such iterative process is quite time-consuming, which partly sacrifices the speed merit of NAT. How to abandon the iterative process while enjoy the benefits of explicitly modeling word dependencies in NAT is still an open problem.

In this paper, we argue that the major culprit of the problem that mask language models have to be used together with iterative inference, is the sampling strategy of masking words in MLM. In particular, MLM employs a fixed uniform strategy for masking words randomly during training, which prevents the model from effectively learning word dependencies for one-iteration generation. For example, at the beginning of training, the NAT model would be poorly tuned and we should mask fewer words. If we mask too many words, it would be difficult for the NAT model to correctly predict the masked words. On the contrary, if we mask too little words at the end phase of training, the resulting NAT model is rarely trained to predict the whole sentences, and can only predict some sentence fragments. In such a case, to accurately generate the whole sentence in inference, the NAT

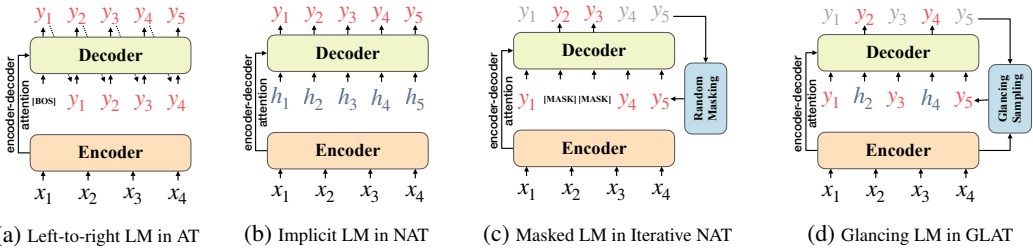

(a) Left-to-right LM in AT  (b) Implicit LM in NAT  (c) Masked LM in Iterative NAT  (d) Glancing LM in GLAT

Figure 1: Different language modeling approaches of different text generation models.

model has to generate the sentence fragments iteratively. To this end, the sampling strategy is crucial for the training of NAT.

To address the above issues, we propose a simple yet effective approach called *Glancing Transformer* (GLAT), which is equipped with the proposed *Glancing Language Model* (GLM) for non-iterative parallel text generation, achieving significant improvements upon strong baselines. Intuitively, GLM adopts a *adaptive glancing sampling* strategy, which glances at some fragments of the reference if the reference is too difficult to fit in the training of NAT. Correspondingly, when the model is well tuned, it will adaptively reduce the percentage of glancing sampling, making sure that the resulting model could learn to generate the whole sentence in a one-iteration fashion.

Specifically, our proposed GLM differs from MLM in two aspects. Firstly, GLM proposes an adaptive glancing sampling strategy, which enables GLAT to generate sentences in a one-iteration way, working by gradual training instead of iterative inference (see Figure 1d). Generally, GLM is quite similar to curriculum learning (Bengio et al., 2009) in spirit, namely first learning to generate some fragments and gradually moving to learn the whole sentences (from easy to hard). To achieve the adaptive glancing sampling, GLM performs decoding twice in training. The *first decoding* is the same as the vanilla NAT, and the prediction accuracy indicates whether current reference is "difficult" for fitting. In the second decoding, GLM gets words of the reference via glancing sampling according to the first decoding, and learn to predict the remaining words that are not sampled. Note that only the second decoding will update the model parameters. Secondly, instead of using the [MASK] token, GLM directly use representations from the encoder at corresponding positions, which is more natural and could enhance the interactions between sampled words and signals from the encoder.

Experimental results show that GLAT obtains significant improvements (about 5 BLEU) on standard benchmarks compared to the vanilla NAT, without losing inference speed-up. GLAT achieves competitive results against iterative approaches like Mask-Predict (Ghazvininejad et al., 2019), even outperforming the Mask-Predict model on WMT14 DE-EN and WMT16 RO-EN. Compared to the strong AT baseline, GLAT can still close the performance gap within 1 BLEU point while keeping $7.9\times$ speed-up. Empirically, we even find that GLAT outperforms AT when the length of the reference is less than 20 on WMT14 DE-EN. We speculate this is because GLM could capture bidirectional context for generation while its left-to-right counterpart is only unidirectional, which indicates the potential of parallel generation approaches like GLAT.

## 2 TEXT GENERATION VIA CONDITIONAL LANGUAGE MODELING

In this section, we compare different language models used in different text generation approaches. Formally, considering a sequence-to-sequence model (Cho et al., 2014; Bahdanau et al., 2014; Vaswani et al., 2017) for predicting $Y = \{y_1, y_2, ..., y_T\}$ given the input sentence $X = \{x_1, x_2, ..., x_N\}$. In the AT model, the training objective is maximizing the log-likelihood with autoregressive decomposition:

$$\mathcal{L}_{\text{AT}} = \sum_{t=1}^{T} \log p(y_t | y_{<t}, X; \theta), \tag{1}$$

where the word $y_t$ is conditioned on the target prefix $y_{<t} = \{[\text{BOS}], y_1, ..., y_{t-1}\}$ and the source input $X$. AT models sentences from left-to-right, therefore word dependencies are learned in a

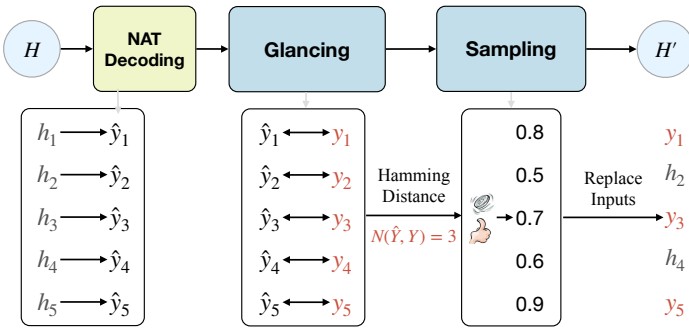

Figure 2: Illustration of the procedure of glancing sampling

unidirectional way. NAT, on the other hand, incorporates conditional independence assumption among words in a sentence with the aim of enabling parallel generation:

$$\mathcal{L}_{\text{NAT}} = \sum_{t=1}^{T} \log p(y_t | X; \theta). \tag{2}$$

Note that, in NAT, different $y_t$ in $Y$ are predicted simultaneously, which removes the interactions among target words in the NAT modeling. Thus, to generate fluent and faithful sentences, NAT has to model the target word dependencies implicitly, which makes the learning process quite challenging.

To explicitly model the target word dependencies, previous iterative approaches, such as Mask-Predict, introduce mask language models (MLM) in NAT. MLM models word dependencies by learning to predict the masked words conditioned on the unmasked ones:

$$\mathcal{L}_{\text{MLM}} = \sum_{y_t \in \mathbb{RM}(Y)} \log p\Big(y_t | \Phi\big(Y, \mathbb{RM}(Y)\big), X; \theta\Big). \tag{3}$$

Here $\mathbb{RM}(Y)$ returns some randomly sampled words from $Y$, and $\Phi$ replaces these sampled words in $Y$ with the [MASK] token. For example, if $\mathbb{RM}(Y) = \{y_2, y_3\}$, $\Phi\big(Y, \mathbb{RM}(Y)\big) = \{y_1, [\text{MASK}], [\text{MASK}], y_4, ...\}$. To this end, the training objective is to predict the masked words $\mathbb{RM}(Y)$ given the source sentence $X$ and the unmasked target words.

As mentioned in the Introduction, though MLM can explicitly model target word dependencies, it can hardly generate satisfactory sentences without iteration. Better language modeling approach should be explored for parallel text generation in the one-iteration way.

## 3 THE PROPOSED GLANCING TRANSFORMER

In this section, we describe GLAT in detail. Generally, GLAT differs from vanilla NAT in that it explicitly models word dependencies via the proposed glancing language model (GLM), which leads to significant accuracy improvements. Additionally, compared to these iterative approaches that adopt MLM, the proposed GLM is also non-trivial for the employment of an adaptive glancing sampling strategy, which enables GLAT to generate promising sentences in a one-iteration way. And GLAT further strengthen the decoder inputs with the source representation.

### 3.1 THE GLANCING LANGUAGE MODEL

Formally, given the training data $D = \{X, Y\}_{i=1}^{L}$, the task is predicting $Y = \{y_1, y_2, ..., y_T\}$ with the input sentence $X = \{x_1, x_2, ..., x_N\}$. By employing GLM, the training objective of GLAT is:

$$\mathcal{L}_{\text{GLAT}} = - \sum_{\{X,Y\} \in D} \sum_{y_t \in \{Y \setminus \mathbb{GS}(Y, \hat{Y})\}} \log p(y_t | \mathbb{GS}(Y, \hat{Y}), X; \theta). \tag{4}$$

Here, $\hat{Y}$ is the predicted sentences in the first decoding pass, and $\mathbb{GS}(Y, \hat{Y})$ is the set of sampled target words by the sampling strategy of $\mathbb{GS}$ (described in detail in the next section). The sampling strategy samples more words from reference $Y$ if prediction $\hat{Y}$ is less accurate, and samples fewer words for the opposite case. Additionally, $\{Y \setminus \mathbb{GS}(Y, \hat{Y})\}$ is the difference set, representing the remaining words except for these sampled words. Then we perform the second decoding, updating the model parameters $\theta$, by maximizing the likelihood of these remaining words with cross-entropy.

Specifically, GLM performs decoding twice in its learning procedure. In the first decoding pass, given the encoder $\mathbb{F}_{\text{encoder}}$ and decoder $\mathbb{F}_{\text{decoder}}$, $H = \{h_1, h_2, ..., h_T\}$ is the encoded output sequence gathered from the input $X$, and $\hat{Y} = \mathbb{F}_{\text{decoder}}(H, \mathbb{F}_{\text{encoder}}(X; \theta); \theta)$ is the predicted sentence in the first decoding pass. With the predicted sentence $\hat{Y}$, $\mathbb{GS}(Y, \hat{Y})$ is the set of sampled words from reference $Y$, according to our adaptive sampling strategy $\mathbb{GS}$ that will be introduced in the next section. Note that we use the attention mechanism to form the decoder inputs with the input $X$. Previous work adopt *Uniform Copy* (Gu et al., 2018) or *SoftCopy* (Wei et al., 2019) instead. But empirically, we find that they produce almost the same results in our setting.

In the second decoding pass, we cover the original decoding inputs $H$ by the embeddings of words from $\mathbb{GS}(Y, \hat{Y})$ to get the new decoding inputs $H' = \mathbb{F}_{\text{cover}}(E_{y_t \in \mathbb{GS}(Y, \hat{Y})}(y_t), H)$, where $\mathbb{F}_{\text{cover}}$ covers according to the corresponding positions. Namely, if we have a sampled word at one position, we use its word embedding to replace the original decoding input at the same position. Here the word embeddings are obtained from the softmax embedding matrix of the decoder. With the mix of encoding signals and reference words from glancing sampling $H'$ as decoder inputs, the training objective of our proposed *glancing language model* can be written as Equation 4, where the probabilities of remaining words on each position $p(y_t \mid \mathbb{GS}(Y, \hat{Y}), X; \theta)$ are computed by $\mathbb{F}_{\text{decoder}}(H', \mathbb{F}_{\text{encoder}}(X; \theta), t; \theta)$.

## 3.2 THE GLANCING SAMPLING STRATEGY

We adopt the glancing sampling in GLM to adaptively sample words from the reference. Intuitively, the sampling strategy of our proposed glancing sampling method will sample many words at the start of the training, when the model is not yet well tuned. As the model gets better progressively, fewer words will be sampled to enable the model to learn the simultaneous generation of the whole sentence. Note that the sampling strategy is crucial in the training of NAT, a good sampling strategy could gradually guide the model to first learn the generation of fragments and then turn to the whole sentences.

As illustrated in Figure 2, the glancing sampling could be divided into two steps: first deciding a sampling number $N$ adaptively, and then *randomly* selecting $N$ words from the reference. The sampling number $N$ will be larger when the model is poorly trained and decreases along the training process. Note that we choose to randomly select the $N$ words from the reference. The random reference word selection is simple and yields good performance empirically.

Formally, given the input $X$, its predicted sentence $\hat{Y}$ and its reference $Y$, the goal of glancing sampling function $\mathbb{GS}(Y, \hat{Y})$ is to obtain a set of sampled words $S$, where $S$ is a subset of $Y$:

$$\mathbb{GS}(Y, \hat{Y}) = \mathcal{R}(Y, N(Y, \hat{Y})) \tag{5}$$

Here, $\mathcal{R}(Y, N(Y, \hat{Y}))$ means randomly selecting $N$ words from $Y$, and $N$ is computed as:

$$N(Y, \hat{Y}) = f_{\text{ratio}} \cdot d(Y, \hat{Y}) \tag{6}$$

where $d(Y, \hat{Y})$ is a metric for measuring the differences between $Y$ and $\hat{Y}$. We adopt the Hamming distance (Hamming, 1950) as the metric, which is computed as $d(Y, \hat{Y}) = \sum_{t=1}^{T}(y_t \neq \hat{y}_t)$. With $d(Y, \hat{Y})$, the sampling number can be decided adaptively considering the training status of the model. Note that here $d(Y, \hat{Y})$ could be other distances such as Levenshtein distance (Levenshtein, 1966), but we find the Hamming distance gives the best result empirically. Additionally, to better control the glancing sampling process, we include a hyper-parameter $f_{\text{ratio}}$ to adjust the number of sampled words more flexibly.

## 4    EXPERIMENTS

In this section, we first introduce the settings of our experiments, then report the main results compared with several strong baselines. Ablation studies and further analysis are also included to verify the effects of different components used in GLAT.

### 4.1    EXPERIMENTAL SETTINGS

**Datasets**    We conduct experiments on three machine translation benchmarks: WMT14 EN-DE (4.5M translation pairs), WMT16 EN-RO (610k translation pairs), and IWSLT16 DE-EN (150K translation pairs). These datasets are tokenized and segmented into subword units using BPE encodings (Sennrich et al., 2016). We preprocess WMT14 EN-DE by following the data preprocessing in Vaswani et al. (2017). For WMT16 EN-RO and IWSLT16 DE-EN, we use the processed data provided in Lee et al. (2018).

**Distillation**    Following previous work (Gu et al., 2018; Lee et al., 2018; Wang et al., 2019), we also use sequence-level knowledge distillation for all datasets. We employ the transformer base Vaswani et al. (2017) as the teacher for knowledge distillation. Then, we train our models on distilled data for each task.

**Inference**    GLAT only modifies the training procedure and performs one-iteration non-autoregressive generation as the vanilla NAT in Gu et al. (2018). Before decoding, GLAT first predict the target length for outputs and the length prediction is implemented as in Ghazvininejad et al. (2019). An additional `[LENGTH]` token is add to the source input, and the encoder output for the `[LENGTH]` token is used to predict the length.

Besides generation with one predicted target length, we also consider the common practice of noise parallel decoding (Gu et al., 2018; Lee et al., 2018; Guo et al., 2019a; Wang et al., 2019), which generates several decoding candidates in parallel and selects the best via re-scoring with a pre-trained autoregressive model. For GLAT, we first predict $m$ target length candidates, then generate output sequences with argmax decoding for each target length candidate. Then we use the pre-trained transformer to rank these sequences and identify the best overall output as the final output.

**Implementation**    We adopt the vanilla model which copies source input uniformly in Gu et al. (2018) as our base model (NAT-base) and replace the *UniformCopy* with attention mechanism using positions. For WMT datasets, we follow the hyperparameters of the base Transformer in Vaswani et al. (2017). And we choose a smaller setting for IWSLT16, considering that IWSLT16 is a smaller dataset. For IWSLT16, we use 5 layers for encoder and decoder and set the model size $d_{model}$ to 256. We train the model with batches of 64k/8k tokens for WMT/IWSLT datasets, respectively. We use Adam optimizer (Kingma & Ba, 2014) with $\beta = (0.9, 0.999)$. For WMT datasets, the learning rate warms up to $5e - 4$ in 4k steps and gradually decays according to inverse square root schedule in Vaswani et al. (2017). As for IWSLT16 DE-EN, we adopt linear annealing (from $3e - 4$ to $1e - 5$) as in Lee et al. (2018). For the hyper-parameter $f_{\text{ratio}}$, we adopt linear annealing from 0.5 to 0.3 for WMT datasets and a fixed value of 0.5 for IWSLT16. The final model is created by averaging the 5 best checkpoints chosen by BLEU scores on the validation set.

**Competitors**    We compare our method with strong representative baselines, including fully non-iterative models: our vanilla NAT-base model, the NAT with fertility (Gu et al., 2018, NAT-FT), the NAT imitating AT (Wei et al., 2019, imit-NAT), the Flow-based NAT (Ma et al., 2019, Flowseq), the NAT with hint-based training (Li et al., 2019, NAT-HINT), Imputer (Saharia et al., 2020), the NAT with CRF (Lafferty et al., 2001) decoding (Sun et al., 2019, NAT-DCRF), and the NAT with iterative refinement: NAR-IR (Lee et al., 2018), LevT (Gu et al., 2019), Mask-Predict (Ghazvininejad et al., 2019), and JM-NAT (Guo et al., 2020). For all our tasks, we obtain other NAT models' performance by directly using the performance figures reported in their papers if they are available on our datasets.

### 4.2    RESULTS

The main results on the benchmarks are presented in Table 1. Obviously, GLAT significantly improves the translation quality and outperforms strong baselines by a large margin. Our method

Table 1: Performance on WMT14 EN-DE/DE-EN and WMT16 EN-RO/RO-EN benchmarks. $I_{dec}$ is the number of decoding iterations and $m$ is the number of reranking candidates.

| Models | | $I_{dec}$ | WMT14 | | WMT16 | | Speed Up |
|---|---|---|---|---|---|---|---|
| | | | EN-DE | DE-EN | EN-RO | RO-EN | |
| AT Models | Transformer (Vaswani) | N | 27.30 | / | / | / | / |
| | Transformer (ours) | N | 27.48 | 31.27 | 33.70 | 34.05 | 1.0× |
| Iterative NAT | NAT-IR | 10 | 21.61 | 25.48 | 29.32 | 30.19 | 1.5× |
| | LevT | 6+ | 27.27 | / | / | 33.26 | 4.0× |
| | Mask-Predict | 10 | 27.03 | 30.53 | **33.08** | **33.31** | / |
| | JM-NAT | 10 | **27.31** | **31.02** | / | / | 5.7× |
| Non-iterative NAT | NAT-FT | 1 | 17.69 | 21.47 | 27.29 | 29.06 | 15.6× |
| | imit-NAT | 1 | 22.44 | 25.67 | 28.61 | 28.90 | 18.6× |
| | NAT-HINT | 1 | 21.11 | 25.24 | / | / | 30.2× |
| | Flowseq | 1 | 23.72 | 28.39 | 29.73 | 30.72 | / |
| | Imputer | 1 | **25.8** | 28.4 | / | / | / |
| | NAT-base (ours) | 1 | 20.36 | 24.81 | 28.47 | 29.43 | 15.3× |
| | GLAT (ours) | 1 | **25.21** | **29.84** | **31.19** | **32.04** | 15.3× |
| Non-iterative NAT w/ Reranking | NAT-FT (m=100) | 1 | 19.17 | 23.20 | 29.79 | 31.44 | 2.4× |
| | imit-NAT (m=7) | 1 | 24.15 | 27.28 | 31.45 | 31.81 | 9.7× |
| | NAT-HINT (m=9) | 1 | 25.20 | 29.52 | / | / | 17.8× |
| | Flowseq (m=30) | 1 | 25.31 | 30.68 | 32.20 | 32.84 | / |
| | NAT-DCRF (m=9) | 1 | 26.07 | 29.68 | / | / | 6.1× |
| | GLAT (m=7, ours) | 1 | **26.55** | **31.02** | **32.87** | **33.51** | 7.9× |

introduces explicit dependency modeling for the decoder and gradually learns simultaneous generation of whole sequences, enabling the model to better capture the underlying data structure. Compared to models with iterative decoding, our method completely maintains the inference efficiency advantage of fully non-autoregressive models, since GLAT generate with only one-iteration. Compared with the competitors, we will highlight our empirical advantages:

- The performance of GLAT is surprisingly good. Compared with the vanilla NAT-base models, GLAT obtains significant improvements (about 5 BLEU) on EN-DE/DE-EN. Additionally, GLAT also outperforms other fully non-autoregressive models with a substantial margin (almost +3 BLEU score on average). The results are even very close to those of the AT model, which shows great potential.

- GLAT is simple and can be applied to other NAT models flexibly, as we only modify the training process by reference glancing while keeping inference unchanged. For comparison, imitate-NAT introduces additional AT models as teachers; NAT-DCRF utilizes CRF to generate sequentially; NAT-IR and Mask-Predict models need multiple decoding iterations.

- Note that Imputer and GLAT use different methods to determine the best target length. Based on CTC (Graves et al., 2006), Imputer sets the max target length twice the length of the source input and determines the best length by removing blanks and contiguous repetitive words after generation. Thus, it is non-trivial to apply target length reranking in Imputer, while GLAT can be further improved from 25.2 to 26.5 with AT reranking on WMT14 EN-DE, which outperforms the Imputer model.

We also present a scatter plot in Figure 3, displaying the trend of speed-up and BLEU scores with different NAT models. It is shown that the point of GLAT is located on the top-right of the competing methods. Obviously, GLAT outperforms our competitors in BLEU if speed-up is controlled, and in speed-up, if BLEU is controlled. This indicates that GLAT outperforms previous state-of-the-art NAT methods. Although iterative models like Mask-Predict achieves competitive BLEU scores, they only maintain minor speed advantages over AT. In contrast, fully non-autoregressive models remarkably improve the inference speed.

### 4.3 ANALYSIS

**Effect of Source Input Length** To analyze the effect of source input length on the models' performance, we split the source sentences into different intervals by length after BPE and compute

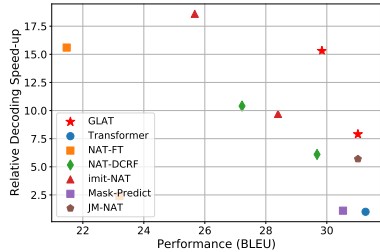
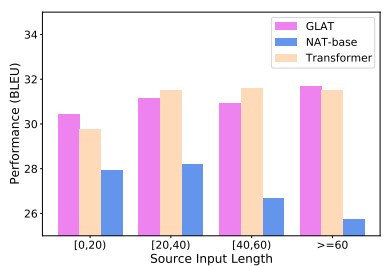

Figure 3: The trade-off between speed-up and BLEU on WMT14 DE-EN

Figure 4: Performance under different source input length on WMT14 DE-EN

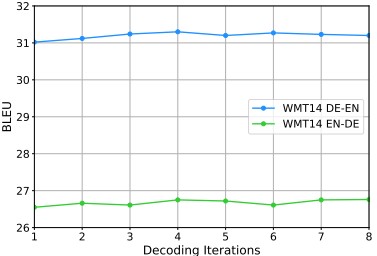
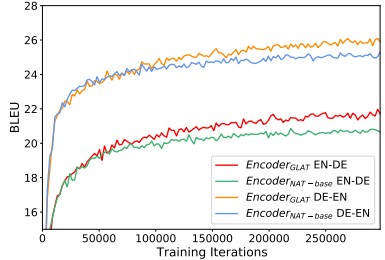

Figure 5: The BLEU scores of GLAT with different decoding iterations

Figure 6: Training NAT with different initialized encoder

the BLEU score for each interval. The histogram of results is presented in Figure 4. NAT-base's performance drops sharply for long sentences, while the gradual learning process enables GLAT to boost the performance by a large margin, especially for long sentences. We also find that GLAT outperforms autoregressive Transformer when the source input length is smaller than 20.

**GLAT Achieves Strong Results without Multiple Iterations**  We conduct experiments of GLAT with more than one decoding iteration in inference. We adopt the inference algorithm in Ghazvininejad et al. (2019) for multiple-iteration decoding. The results are shown in Figure 5. We find that GLAT can achieve decent performances with only one decoding iteration, while further iterations only obtain minor improvements of 0.2∼0.3 BLEU.

**GLAT Learns Better Encoder**  To validate whether GLAT strengthens encoders, we train NAT models with different initialization for comparison. We choose NAT-base and GLAT for comparison and the only difference between them is the training procedure. Specifically, we initialize the encoders of NAT models with the parameter of trained GLAT and trained NAT-base, respectively. The parameters of encoders are fixed during training. The models are all trained using the traditional MLE loss, and the results are presented in Figure 6. Obviously, models initialized with encoders of GLAT outperform those with NAT-base encoders, indicating that GLAT learns better encoders.

**Training Speed of GLAT**  Training on 8 V100 GPUs for WMT14 EN-DE, NAT-base takes about 45 hours and GLAT takes about 56 hours. Compared to NAT-base, the training of GLAT is about 1.2 times slower. Because we only add one forward pass of the decoder, the overhead is relatively small.

### 4.4 ABLATION STUDY

**Effectiveness of the Adaptive Sampling Number**  To validate the effectiveness of the adaptive sampling strategy for the sampling number $N(Y, \hat{Y})$, we also introduce two fixed approaches for comparison. The first one decides the sampling number with $\lambda * T$, where $T$ is the length of $Y$, and $\lambda$ is a constant ratio. The second one is relatively flexible, which sets a start ratio of $\lambda_s$ and a end ratio $\lambda_e$, and linearly reduce the sampling number from $\lambda_s * T$ to $\lambda_e * T$ with the training process.

Table 2: Performances on IWSLT16 with fixed sampling ratio.

| Sampling Method | $\lambda$ | BLEU |
|---|---|---|
| Fixed | 0.0 | 24.66 |
| | 0.1 | 24.91 |
| | 0.2 | 27.12 |
| | 0.3 | 24.98 |
| | 0.4 | 22.96 |
| Adaptive | - | **29.61** |

Table 3: Performances on IWSLT16 with decreasing sampling ratio.

| Sampling Method | Schedule | | BLEU |
|---|---|---|---|
| | $\lambda_s$ | $\lambda_e$ | |
| Fixed | 0.5 | 0 | 27.80 |
| | 0.5 | 0.1 | 28.21 |
| | 0.5 | 0.2 | 27.15 |
| | 0.5 | 0.3 | 23.37 |
| Adaptive | - | | **29.61** |

Table 4: Performance on WMT14 EN-DE with different reference word selection strategies.

| Selection Strategy | random | $p_{\text{ref}}$ | $1 - p_{\text{ref}}$ | most certain | most uncertain |
|---|---|---|---|---|---|
| GLAT | 25.21 | 24.87 | 25.37 | 24.99 | 24.86 |
| GLAT (w/ reranking m=7) | 26.55 | 25.83 | 26.52 | 26.22 | 26.13 |

Table 5: Ablation study on WMT14 EN-DE and WMT14 DE-EN.

| | WMT14 EN-DE | WMT14 DE-EN |
|---|---|---|
| GLAT w/ sampling strategy of Mask-Predict | 19.16 | 23.56 |
| GLAT w/ decoder inputs of Mask-Predict | 24.99 | 29.48 |
| GLAT | 25.21 | 29.84 |

As shown in Table 2 and Table 3, clearly, our adaptive approach (Adaptive in the table) outperforms the competitors with big margins. The results confirm our intuition that the sampling schedule affects the generation performance of our NAT model. The sampling strategy, which first offers relatively easy generation problems and then turns harder, benefits the final performance. Besides, even with the simplest constant ratio, GLAT still achieves remarkable results. When set $\lambda = 0.2$, it even outperforms the baseline $\lambda = 0.0$ by 2.5 BLEU score.

The experiments potentially support that it is beneficial to learn the generation of fragments at the start and gradually transfer to the whole sequence. The flexible decreasing ratio method works better than the constant one, and our proposed adaptive approaches achieve the best results.

**Influence of Reference Word Selection** To analyze how the strategies of selecting reference words affect in glancing sampling, we conduct experiments with different selection strategies. By default, we assume all the words in the reference are equally important and randomly choose reference words for glancing. Besides the random strategy, we devise four other selection methods considering the prediction of first decoding. For $p_{\text{ref}}$ and $1 - p_{\text{ref}}$, the sampling probability of each reference word is proportional to the output probability for the reference word $p_{\text{ref}}$ or $1 - p_{\text{ref}}$, respectively. Similar to the word selection strategy for masking words during inference in Mask-Predict, we also add two strategies related to the prediction confidence: "most certain" and "most uncertain". We choose the positions where predictions have higher confidence for "most certain", and vise versa for "most uncertain". The results for different selection methods are listed in Table 4.

In comparisons, the model with the selection strategy $1 - p_{\text{ref}}$ outperforms the one with $p_{\text{ref}}$, indicating that words hard to predict are more important for glancing in the training process. And we find that the random strategy performs a little better than the two confidence-based strategies. We think this indicates that introducing more randomness in the glancing sampling could enable GLAT to learn to explore more dependencies among target words. We adopt the random strategy for its simplicity and good performance.

**Advantages of GLAT over Mask-Predict** To study the effects of sampling strategy and decoder inputs of GLAT, we report the results of replacing these two modules in GLAT with the corresponding

Table 6: Performance on WMT14 EN-DE and WMT14 DE-EN with different distances.

| | WMT14 EN-DE | | WMT14 DE-EN | |
|---|---|---|---|---|
| | Hamming | Levenshtein | Hamming | Levenshtein |
| GLAT | 25.21 | 24.56 | 29.84 | 28.96 |
| GLAT (w/ reranking m=7) | 26.55 | 26.21 | 31.02 | 30.85 |

part in Mask-Predict, respectively in Table 5. GLAT employs glancing sampling strategy instead of the uniform sampling strategy used in Mask-Predict, and replaces the [MASK] token with source representation from the encoder. The results show that the glancing sampling strategy outperforms the uniform sampling strategy by 5∼6 BLEU scores, and feeding representations from the encoder as the decoder input could still improve the strong baseline by 0.2∼0.3 BLEU scores after adopting glancing sampling. To sum up, the adaptive glancing sampling approach contributes the most to the final improvement, and the use of representations from the encoder also helps a bit.

**Comparison of Different Distances for Glancing Sampling**  We conduct experiments with two distances for comparing the predictions of first decoding and references, and the results are presented in Table 6. Experimental results show that both distances can be used to improve the quality of one-iteration generation, and GLAT with Hamming distance is better than GLAT with Levenshtein distance (Levenshtein, 1966). We think Hamming distance is more strict than Levenshtein distance because only the same words on the corresponding positions are regarded as correct, which is more consistent with the training of GLAT.

## 5 RELATED WORK

**Fully Non-Autoregressive Models**  A line of work introduces various forms of latent variables to reduce the model's burden of dealing with dependencies among output words (Gu et al., 2018; Ma et al., 2019; Bao et al., 2019; Ran et al., 2019). Another branch of work considers transferring the knowledge from autoregressive models to non-autoregressive models (Wei et al., 2019; Li et al., 2019; Guo et al., 2019b). Besides, there are also some work that apply different training objectives to train non-autoregressive models (Libovickỳ & Helcl, 2018; Shao et al., 2020; Ghazvininejad et al., 2020) or add regularization terms (Wang et al., 2019; Guo et al., 2019a).

**Non-Autoregressive Models with Structured Decoding**  To model the dependencies between words, Sun et al. (2019) introduces a CRF inference module in NAT and performs additional sequential decoding after the non-autoregressive computation in inference. Deng & Rush (2020) proposes cascaded CRF decoding. Since GLAT only performs one-iteration non-autoregressive generation, our approach is orthogonal to the method proposed in Sun et al. (2019). We can also combine our approach with the structured decoding method.

**Non-Autoregressive Models with Iterative Refinement**  A series of work are devotes to semi-autoregressive models that combine the strength of both types of models by iteratively refining the former outputs. Lee et al. (2018) proposed a method of iterative refinement based on denoising autoencoder. Gu et al. (2019) utilized insertion and deletion to refine the generation. Ghazvininejad et al. (2019) trained the model in the way of the masked language model, and the model iteratively replaces the mask tokens with new outputs. Despite the relatively better accuracy, the multiple decoding iterations vastly reduces the inference efficiency of non-autoregressive models.

## 6 CONCLUSION

In this paper, we propose Glancing Transformer with a glancing language model to improve the performance of non-iterative NAT. With the glancing language model, the model reduces learning difficulty by gradually learning with growing targets for simultaneous generation and explicitly models target word dependencies. Experimental results show that our approach significantly improves the performance of non-autoregressive machine translation with one-iteration generation.

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
