# OpenReview forum: "Non-iterative Parallel Text Generation via Glancing Transformer"
_ICLR.cc/2021/Conference — Reject_

### Official Review · AnonReviewer3 · 2020-10-28
**curriculum learning based technique to improve single step parallel generation for MT**

**Rating:** 5
**Confidence:** 4

**Review:**

The authors propose Glancing Transformer for single step parallel text generation. The approach is inspired from curriculum learning i.e. the training task is adaptively controlled based on the model's current performance. Specifically, the paper proposes a glancing strategy which compares the model's generation and reference sentence, and forms a sequence which is partially masked. The number of masked tokens in this sequence depends on the similarity between model's generation and reference sentence. The model is then trained to complete the partially masked sequence.

The model achieves strong improvements on standard non-autoregressive MT baselines while not modifying inference process, thus not compromising on inference speed over vanilla non-autoregressive models.

+ves :
- Limitations of current non-autoregressive MT models (NAT) are well explained, and the approach is nicely motivated. The paper is well written (although there are many grammatical mistakes that can be revised by the authors) and is easy to follow. The experimental details are well documented, many ablation studies are reported apart from comparison using standard metrics.

- The results on standard benchmarks are strong. The paper improves over vanilla NAT by approx. 5 BLEU points on average, and is only 1 BLEU point less than baseline autoregressive models while being ~7x faster in inference.

Concerns :

- While the use of curriculum learning inspired techniques to augment NAT model training is new and interesting, their specific technique does not seem to have sufficient novelty according to me. If I understood their technique correctly, the only difference in the training algorithm between Mask-Predict (Ghazvininejad et al., 2019) and their method is the selection of "number of tokens to mask (`mask-num`)" in the decoder's input. While Mask-Predict uses a uniform distribution to sample `mask-num` , they use a glancing sampler that decides this number based on the hamming distance between model's prediction and reference sentence.

- Although the reported results and ablation studies show a significant impact of this simple change, I think more exploration of this technique is possible and should be done in the paper. E.g. A simple hamming distance may not be a good strategy to compare the model's prediction and reference sentence. A related finding has been described in detail by Ghazvininejad et al., 2020 (https://arxiv.org/abs/2004.01655). So, I believe that authors can explore more strategies to compare reference and generated sequence.

- The authors use Random Sampling like Mask-Predict as their sampling strategy. The authors argue that random sampling may not be the most efficient strategy, but it's the easiest one, and has been shown to be powerful in models like BERT. While I totally agree with their argument, I believe that there is some possibility here to exploit the fact that we have access to model's prediction and reference sentence.  E.g. one possible strategy to exploit this could be selecting tokens that the model was not able to predict correctly based on the hamming distance comparison.

- In the Introduction section, it is mentioned that - "Second, GLM replaces the [MASK] token in the MLM with signals from the encoder, thus not only
the target word dependency, but also the dependencies between the input and target are strengthened." While this is an interesting argument, I didn't find any experiment to validate this. I think the authors should include experiments to compare use of [MASK] tokens and their approach empirically.

Based on these concerns, I am currently inclined to recommend rejection. While I find the idea of incorporating curriculum learning very interesting (together with strong results as demonstrated by the paper), I believe that more exploration of strategies to sample number of masked tokens, and sample words from the reference sentence is necessary to make the paper publishable. I have described this point in more detail in the Concerns above.

Minor Comments :
- There are many grammatical errors in the current version of the paper that the authors might want to revise. (E.g. in abstract - falls -> fall, achieves -> achieve).
- In Section 3.1 authors mention this - "Note that we use the attention mechanism to form the decoder inputs with the input X." I think it might be helpful to elaborate more on what this exactly means.

---

> ### Author Response · Authors · 2020-11-23
> **We have compared more strategies for the sampling number of reference words, and the selection of words sampled from references**
>
> Thanks for your constructive suggestions.
> To address your concern that *"I believe that more exploration of strategies to sample number of masked tokens, and sample words from the reference sentence is necessary to make the paper publishable."*, we added several results to compare more strategies of sampling number and more strategies of selecting/sampling words from the reference sentence: a) besides the default random reference selection strategy, we add four other strategies that leverage the prediction of first decoding; b) besides the Hamming distance, we include the Levenshtein distance in GLAT for comparison. And we also added ablation studies for comparisons between GLAT and Mask-Predict.
>
> **Question1: Novelty compared with mask-predict**
>
> Thanks for your kind notes. Intuitively, GLAT is quite different from Mask-Predict in following aspects:
> 1) GLAT is a non-iterative generation model while Mask-Predict relies on multiple decoding iterations.
> 2) The difference between GLAT and Mask-Predict seems to mainly lie in the sampling strategy, but the insight behind is quite different. The initial motivation of GLAT is to boost the performance of one-iteration generation. We find that explicit word dependency modeling is crucial for NAT, while the language model for one-iteration generation has not been well designed. Besides, we also find the training of NAT is very steep, and we finally turn to the idea of curriculum learning by first learning the generation of fragments and gradually moving to whole sequences. However, intuitions of Mask-Predict come from different ways, they want to train an iterative editing model. They find mask language model is suitable for leveraging partially generated outputs, and finally apply MLM in the iterative editing setting, which gives very promising results.
> 3) Although the adaptive glancing sampling is very simple, it is very effective, which leads to significant improvements compared to uniform sampling (about 6 BLEU on WMT EN-DE & DE-EN).
> We hope the above reply could address your concerns about the novelty of GLAT, and any further comments/questions are welcome!
>
> **Question2: Experiments with more sampling strategies. E.g. One possible strategy to exploit this could be selecting tokens that the model was not able to predict correctly based on the hamming distance comparison.**
>
> Thank you for the constructive advice. Besides the default random sampling strategy for reference word selection, we add two strategies related to the prediction probability of the words in reference: $p_\text{ref}$ and $1-p_\text{ref}$. Note that the two sampling approaches are used after the adaptive sampling number N is obtained.
> For $p_\text{ref}$, the sampling probability of each reference word is proportional to the word output probability $p_\text{ref}$ on the corresponding position. For $1-p_\text{ref}$, the sampling probability is proportional to $1-p_\text{ref}$.
> Similar to the word selection strategy for masking words during inference in Mask-Predict, we also add two strategies related to the prediction confidence: "most certain", "most uncertain". For "most certain", we sample words at these positions with higher prediction confidence. For "most uncertain", we choose those with the lower prediction confidence.
> The results are list as follows:
>
> |		|	|random	||	$p_\text{ref}$	||	$1-p_\text{ref}$||	most certain	||	most uncertain|
> |-----------|---|:-----------:|---------------|:----------------:||:---------:||:------------:||:-----------------:|
> |GLAT	|	|25.21	||	24.87	||	25.37		|| 24.99		||	    24.86	|
> |GLAT(with reranking)|	|26.55	||	25.83||		26.52	||	26.22	||	26.13|
> |
>
> The results show that GLAT can significantly improve the performance with all these strategies for reference word selection.
> In comparisons, the model with the selection strategy $1-p_\text{ref}$ outperforms the one with $p_\text{ref}$, indicating that words hard to predict are more important for glancing in the training process.
> And we find that the random strategy performs a little better than the two confidence-based strategies. We think this indicates that introducing more randomness in the glancing sampling could enable GLAT to learn to explore more dependencies among target words. We adopt the random strategy for its simplicity and good performance.

---

> > ### Author Response · Authors · 2020-11-23
> > **We have compared more strategies for the sampling number of reference words, and the selection of words sampled from references (2)**
> >
> > **Question3: Experiments with more distances for comparing the model's prediction and reference sentence**
> >
> > Thanks, we add experiments of GLAT with Levenshtein distance and list the results of both Hamming distance and Levenshtein distance as follow:
> >
> > |			|	|WMT14 EN-DE|			|	|	WMT14 DE-EN|				|
> > |---------------|-----|:--------------|:-------------------|---|:---------------------|:----------------|
> > |			|	|Hamming	|	Levenshtein	|	|	Hamming	|  Levenshtein|
> > |GLAT		|	|25.21		|	24.56		|	|	29.84		|	28.96	|
> > |GLAT(with reranking)|	|26.55	|		26.21	|	|		31.02	|		30.85|
> > |
> >
> > Both distances can be used in GLAT for obtaining improvements by one-iteration generation and we find that the GLAT with Hamming distance performs better than the Levenshtein counterpart. Intuitively, the Hamming distance is a more strict metric compared to the Levenshtein distance. Specifically, only correctly predicting the same words at the corresponding position of the reference is regarded as matching with the reference (the Levenshtein distance does not punish position errors a lot).  Note that such a strict distance metric is more consistent with the MLE objective used in the training of GLAT.
> >
> > **Question4:  I think the authors should include experiments to compare use of [MASK] tokens and their approach empirically.**
> >
> > Thanks for your suggestion. We did a quick run and compare our proposed GLAT with Mask-Predict by:
> > 1. We replace the glancing sampling with the uniform random sampling used by mask-predict
> > 2. We use the [MASK] token as our decoder inputs but keep all other aspects of training the same
> >
> > |										|   |WMT14 EN-DE	|	WMT14 DE-EN |
> > |---------------------------------------------------|---|:----------------------:|:---------------------:|
> > |GLAT w/ sampling strategy of Mask-Predict	|	|	19.16		|		23.56	|
> > |GLAT w/ decoder inputs of Mask-Predict 	|	|	24.99		|		29.48	|
> > |GLAT									|	|	25.21		|		29.84	|
> > |
> >
> > The results show:
> > 1. the proposed glancing  sampling is crucial, without which the results will drop significantly (5 ~ 6 BLEU scores)
> > 2. Replacing the decoder inputs with the MASK token will also lead to a small drop (0.2 ~ 0.3 BLEU). Note that although the performance drop is not significant, feeding representations from the encoder as the decoder input could still moderately improve the strong baseline after adopting glancing sampling (23.56 ->29.48 -> 29.84 on DE-EN)
> >
> > To sum up, the adaptive glancing sampling approach contributes the most to the final improvement and the use of the representations from the encoder also helps a bit.
> >
> > **Question5: How is attentional copy implemented**
> >
> > We use the positional embedding of the decoder as the query in the attention mechanism and the source representation as the key and the value. With the positional embedding and the source representation, we can compute a distribution over the source representation and the result of attention is the weighted sum of source representation with the distribution.
> >
> > **Question6: Grammatical errors**
> >
> > Thanks, we fixed grammatical errors and polished the paper in the revised submission.

---

### Official Review · AnonReviewer2 · 2020-10-28
**Promising results, unsatisfactory delivery**

**Rating:** 5
**Confidence:** 4

**Review:**

This paper proposes a non-autoregressive neural machine translation model that does not require multiple iterations to achieve a good translation quality. The key difference to previous models is that during training it uses decoding to estimate the number of words to randomly sample (proportionally to the error) as opposed to random sampling a fixed number of them and uses this sampling strategy to mask tokens.

**Strengths**

The idea of using the model to come up with the number of words to sample before predicting them looks new and it appears to be helping a lot the model to do better with a single iteration.

The evaluation shows that the proposed model outperforms previous non-iterative ones and performs on par with iterative ones in some cases while being significantly faster which is promising (but it still relies on reranking based on an autoregressive model).

Moreover, the method performs much better than the non-autoregressive baseline on longer sequences (a trait that is observed in autoregressive models) and performs better than the autoregressive baseline on very short sequences (up to 20 tokens).

**Weaknesses**

(1) The writing requires some more effort because it has some grammatical/spelling errors and was not clear in several parts. It would help simplify wording and make more clear statements. For instance, there is a lot of time spent describing the glancing sampling strategy while it could be described in one paragraph (random sampling strategy paragraph seems redundant).

(2) The proposed idea has some merits and works well judging from the results but it seems somewhat incremental and not very clearly explained (see below). The connection to curriculum learning was a bit hand-wavy and hard to follow. The model does not seem to be trained on targets that are of increasing difficulty but rather the number of incorrect targets simply defines the number of *random* samples to be used for training. Was this the intended connection to this line of work? Having many errors in the prediction during training does not necessarily mean that the example is difficult.

(3) In Section 3.3, how is the function f_{ratio} actually implemented and trained (if applicable)? It's not clear from the provided description of how this is done. How are the gradients computed through the sampling process if it's trainable? This is an essential component of the model and it hasn't been explained well.

(4) Another setback for the reader is the lack of discussion or acknowledgment of the computational cost that is required by the method during training. Performing decoding twice during training looks interesting from the modeling perspective but what is its effect on training speed? The evaluation focus is mainly on inference time but the training speed factor should also play a role too when deciding which method to use.

(5) Related to the above, the non-autoregressive models rely on knowledge distillation and re-ranking based on a pretrained autoregressive model. This already has its own training cost, so increasing it more could lead to a situation where the benefits during inference are overshadowed by the computational cost from training.


Other comments:

Are the models trained until convergence or for a fixed number of steps?  I am wondering what is the impact of the glancing sampling strategy on the convergence.

How did the authors come up with the number of reranking candidates for each model? It looks like the number is different for each model; this should affect the quality of the model.

Could the authors elaborate on what do they mean by "the formulation of our proposed glancing language model *could* be maximizing the likelihood of remaining words..."? Does it actually do that? This was unclear and it leads the reader to make guesses.

In Section 4.2, what do you mean by "BLEU and speed-up are more or less contradictory"?

In Figure 1 and 2, the replaced inputs use the notation h_2, h_3 which points to the encoded inputs but in the textual description the replaced tokens seemed to be coming from the embedding on the decoder side (Section 3.2, paragraph 2): E_{y_t \in GS(Y,\hat{Y} }(y_t). There seems to be this inconsistency between the notation in the diagrams and the notation in text, which is very confusing.

---

> ### Author Response · Authors · 2020-11-23
> **We have improved the delivery and given more discussions as requested (1)**
>
> Thanks very much for your comments. We will first clarify the connection between our method and curriculum learning, then give discussions about the training speed, and finally answer questions about the detailed implementation of GLAT.
>
> **Question1: Having many errors in the prediction during training does not necessarily mean that the example is difficult, the connection with curriculum learning**
>
> Actually, the "difficulty" mentioned in our paper means hard to fit in training, and such "difficulty" can be in proportion to the prediction errors in the first decoding pass of training.
> The intuitive connection between GLAT and curriculum learning is that it is very hard to learn "difficult" data points in the initial phase of training while we can make the model learn these "less difficult" data points. Along with the training process, the model gets better tuned and then turns to learn these "difficult" data points. Such an idea of curriculum learning could guide the model to learn better parameters in training.
> Specifically, to reduce the difficulty of learning,  GLAT samples more target words as decoder inputs and the model learns to fit the remaining fragments when the model is not well tuned. As the model is trained better along the training process, the model glances fewer target words and gradually learns to fit the whole sequence. We have made it clearer in our revised submission (See Section 1).
>
> **Question2: Performing decoding twice during training looks interesting from the modeling perspective but what is its effect on training speed?**
>
> Thanks for pointing out this. We have added the discussion for training speed in the updated submission (See Section 4.3). By training on the same device (8 V100 GPUs) for WMT14 EN-DE, NAT-base takes about 45 hours while GLAT takes about 56 hours. Compared to NAT-base, the training of GLAT is about 1.2 times slower, which is acceptable for achieving significant accuracy improvements.
>
> **Question3: Reranking and KD already have their own training cost, increasing it more could lead to a situation where the benefits during inference are overshadowed by the computational cost from training.**
>
> Yes, KD indeed introduces extra training costs, however, we mainly focus on accelerating the inference speeds of text generation, which is also the main purpose of most NAT works.  Reducing the training cost introduced by KD is not in the scope of this work.
> Additionally, as mentioned in last answer, we think the 1.2 times slower training speed is acceptable considering the significant improvement.
>
> **Question4: Writing: grammatical errors and simplification**
>
> Thanks for your kind notes, we have fixed the grammatical errors and simplified the redundant parts in our updated paper. Especially, we have rephrased the description of the glancing sampling strategy section (Please see Section 3.2).
>
> **Question5: How is the function f_{ratio} actually implemented and trained (if applicable)?**
>
> Sorry for the misleading. We have rephrased the description of f_{ratio} in Section 4.1 to make it clear. The ratio function is a hyper-parameter for better controlling the sampling number, which is not trainable. We adopt linear annealing from 0.5 to 0.3 for WMT datasets and a fixed value of 0.5 for IWSLT16.
>
> **Question6: Are the models trained until convergence or for a fixed number of steps?**
>
> We train the models for 300K steps, which is the same as in Mask-Predict.
> Regarding the impact of the glancing sampling strategy on the convergence, we find that NAT-base and GLAT give similar converge trends in practice, which indicates that the glancing sampling does not slow down the convergence of GLAT while predicting partial sentences.
>
> **Question7: How did the authors come up with the number of reranking candidates for each model?**
>
> Because both 9 candidates and 7 candidates had been used for target length reranking in previous work, all the results of GLAT with reranking are reported by using 7 reranking candidates for fair comparisons. But empirically, we find  results of  7  and  9 reranking candidates are similar. The reranking results with 7 and 9 candidates are:
>
> |                   | EN-DE | DE-EN | EN-RO | RO-EN |
> |----------------------------|:-------:|:-------:|:-------:|:-------:|
> |GLAT(reranking 7 candidates)| 26.55 | 31.02 | 32.87 | 33.51 |
> |GLAT(reranking 9 candidates)| 26.67 | 31.13 | 33.11 | 33.57 |
> ||

---

> > ### Author Response · Authors · 2020-11-23
> > **We have improved the delivery and given more discussions as requested (2)**
> >
> > **Question8: Could the authors elaborate on what do they mean by "the formulation of our proposed glancing language model could be maximizing the likelihood of remaining words..."? Does it actually do that?**
> >
> > Sorry for the confusion, we should replace "could be" with "is" here. We have fixed it in the paper. Thanks for your kind reminder.
> >
> > **Question9: In Figure 1 and 2, the replaced inputs use the notation h_2, h_4 which points to the encoded inputs but in the textual description the replaced tokens seemed to be coming from the embedding on the decoder side (Section 3.2, paragraph 2): E_{y_t \in GS(Y,\hat{Y} }(y_t).**
> >
> > The replaced inputs in GLAT are word embeddings of sampled words by glancing sampling, here "replace" refers to replacing the original encoded inputs with such word embeddings at responding positions. Specifically, the notation h_2, h_4 represent copied representations from source rather than the replaced tokens, and the replaced tokens from the target embeddings are represented by y_1, y_3 and y_5.
> > We think the misunderstandings here may be caused by the symbol colors used in Figure 1 and Figure 2, we have modified the two figures to avoid misunderstanding.

---

### Official Review · AnonReviewer1 · 2020-11-01
**Inspiring idea for training "one-iteration" NAT**

**Rating:** 7
**Confidence:** 5

**Review:**

- Overall Comment
This submission improves non-autoregressive translation (NAT) by proposing a non-iterative parallel text generation model called Glancing Transformer (GLAT), which includes the explicit word dependency modeling in NAT via a proposed Glancing Language Model (GLM). Compared to previous work, biggest contribution of the proposed method is that it improves the training of NAT model with a similar idea of curriculum learning, while keeping the inference time unchanged, setting a significant improvement for non-iterative NAT without reranking. It would be a good baseline for future research on non-iterative NAT models.
However, I still have the following comments and questions:
- Methods
(1) Why wouldn't the model gets stuck at predicting only part of the words correctly?
    As described by the algorithm, the model will sample more reference words as the inputs of the decoder if the prediction is incorrect. However, the loss is only calculated for the remaining words.
Will the model only learn to predict easy words and give up learning the difficult words? I guess the random sampling strategy might be the key but it would be good to know answers from the authors.
(2) Since the model only updates in the second pass where the inputs are always mixed with reference words and source embeddings. There will be a clear mismatch between the second pass and the first pass. Why can the first model be sure to improve while training the second pass? Only by sharing parameters between two decoding passes?
(3) Hamming distance is quite weak. Will the proposed method also apply to other distance such as Levenshtein distance or a learnable distance?
(4) What do you mean by "ratio function" $f_\text{ratio}$? What is the input to this function? Steps?
- Experiments
(1) The proposed training process is in fact very similar to Mask-Predict except the inputs (Encoder hidden states instead of [MASK]) and adjusting the number of reference during training.  It would be nice to have a fair comparison with Mask-Predict for at least two settings (and combined):
(A) Mask-Predict with encoder hidden state inputs (with uniform copy or attention)
(B) Mask-Predict with 1 decoding iteration.
(2) I did not see how the model handle the length. With another network for length prediction? Or decoding with multiple lengths? For the latter case, how to decide these lengths?
- Missing Reference
This paper had an even higher one-iteration NAT results (**25.8** with Imputer compared to **25.21** in the submission on WMT En-De). Although the methods are different and the difference is to be honest marginal, it is important to include discussions on that or combine them in future work.
 *Saharia, Chitwan, et al. "Non-autoregressive machine translation with latent alignments." arXiv preprint arXiv:2004.07437 (2020).*

---

> ### Author Response · Authors · 2020-11-23
> **We have added more discussions about the method and more comparisons with Mask-Predict (1)**
>
> Thanks for your positive and insightful comments. We have added more discussions about a) why GLAT does not get stuck at predicting only parts of words, and b) the mismatch between two decoding passes in training. We also provided more experiments for comparisons: a) ablation studies for comparisons between GLAT and Mask-Predict; b) besides the hamming distance, we include the Levenshtein distance in GLAT for comparison.
> Additionally, we also clarify the details of ratio function and length prediction, as well as provide discussions about the Imputer model.
>
> **Question 1: Why wouldn't the model get stuck at predicting only part of the words correctly? Will the model only learn to predict easy words and give up learning the difficult words? I guess the random sampling strategy might be the key but it would be good to know answers from the authors.**
>
> Yes, the randomness introduced by the random sampling strategy enables GLAT not to be stuck in training. With such a random sampling strategy, all the words in the target sentence have equal chances to  be set as the learning target. Additionally, GLAT obtains extra randomness in training, because the model gets different batch samples during training, which makes the model parameters keep varying along with the training phase. This prevents GLAT from getting stuck at predicting only parts of the words.
>
> **Question2: There will be a clear mismatch between the second pass and the first pass. Why can the first model be sure to improve while training the second pass? Only by sharing parameters between two decoding passes?**
>
> Thanks for your question. Because the first pass shares a lot of parts with the second pass, not only the parameters but also the encoder inputs and part of the decoder inputs, the first pass could still be improved though there is a mismatch between the second pass and the first pass.
> Additionally, the mismatch will be reduced along with the training process. Because during the training process, the model will be better tuned and the adaptive glancing sampling will glance at fewer and fewer target words. As a result, such a mismatch will be addressed to some extent.
>
> **Question3: Will the proposed method also apply to other distance such as Levenshtein distances or a learnable distance?**
>
> Thanks for your kind suggestions, we add experiments of GLAT with Levenshtein distance and list the results of both Hamming distance and Levenshtein distance as follows:
>
> |			|	|WMT14 EN-DE	    |		|	|	WMT14 DE-EN	|  |
> |-----------------|-----|:--------------|:-------------------|---|:---------------------|:----------------|
> |			|	|Hamming	    |	Levenshtein	|	|	Hamming	|  Levenshtein|
> |GLAT		|	|25.21		|	24.56		|	|	29.84		|	28.96	|
> |GLAT(with reranking)|	|26.55	|	26.21      |	|		31.02	|		30.85|
> |
>
> Both distances can be used in GLAT for obtaining improvements by one-iteration generation and we find that the GLAT with Hamming distance performs better than the Levenshtein counterpart. Intuitively, the Hamming distance is a more strict metric compared to the Levenshtein distance. Specifically, only correctly predicting the same words at the corresponding position of the reference is regarded as matching with the reference (the Levenshtein distance does not punish position errors a lot).  Note that such a strict distance metric is more consistent with the MLE objective used in the training of GLAT.
>
> **Question4: Ablation study compared with mask-predict**
>
> Thanks for pointing out this. We did a quick run and compare our proposed GLAT with Mask-Predict by:
> 1. We replace the glancing sampling with the uniform random sampling used by mask-predict
> 2. We use the MASK token as our decoder inputs but keep all other aspects of training the same
>
> |										|   |WMT14 EN-DE	|	WMT14 DE-EN |
> |---------------------------------------------------|---|:----------------------:|:---------------------:|
> |GLAT w/ sampling strategy of Mask-Predict	|	|	19.16		|		23.56	|
> |GLAT w/ decoder inputs of Mask-Predict 	|	|	24.99		|		29.48	|
> |GLAT									|	|	25.21		|		29.84	|
> |
>
> The results show:
> 1. the proposed glancing  sampling is crucial, without which the results will drop significantly (5 ~ 6 BLEU scores)
> 2. Replacing the decoder inputs with the MASK token will also lead to a small drop (0.2 ~ 0.3 BLEU). Although the performance drop is marginal, feeding representations from the encoder as the decoder input could still moderately improve the strong baseline after adopting the glancing sampling (23.56 ->29.48 -> 29.84 on DE-EN)
>
> To sum up, the adaptive glancing sampling approach contributes the most to the final improvement of GLAT, while the use of representations from the encoder also helps a bit.

---

> > ### Author Response · Authors · 2020-11-23
> > **We have added more discussions about the method and more comparisons with Mask-Predict (2)**
> >
> > **Question5: What do you mean by "ratio function" ? What is the input to this function? Steps?**
> >
> > Sorry for the confusion with "ratio function". The ratio function is a hyper-parameter for better controlling the sampling number, and the input to this function is the training step. We adopt linear annealing from 0.5 to 0.3 for WMT datasets and a fixed value of 0.5 for IWSLT16. We have made it clear in the revised submission (Please see Section 4.1)
> >
> > **Question6: How does the model handle the length?**
> >
> > We follow the implementation in Mask-predict for length prediction. Similar to the [CLS] token in BERT(Devlin et al., 2019), a special [LENGTH] token is added to the source input of the encoder. And the encoder output of the additional [LENGTH] token is used to predict the target length for the decoder.
> > We have clarified this in our updated version of our paper (Section 4.1).
> >
> > **Question7: Add discussion and comparison with Imputer**
> >
> > Thanks, we added the discussion with Imputer in our updated paper. Imputer and GLAT are orthogonal approaches for boosting the performances of NAT without iterative inference.
> >
> > Additionally, the two models use different methods to determine the best target length. Based on CTC (Graves et al., 2016), Imputer sets the max target length twice the length of source input and determines the best length by removing blanks and contiguous repetitive words after generation. Thus,  it is non-trivial to apply the length reranking trick in Imputer, while GLAT can be further improved from 25.2 to 26.5 with AT reranking (Note that GLAT can also employ itself instead of AT in length reranking, still achives 26.0 ) on WMT EN-DE, which outperforms the Imputer model.
> >
> > Besides, GLAT and CTC use quite orthogonal techniques to improve generation quality,  thus the combination of them would also be a promising direction for future work. GLAT starts from learning to generate sequence fragments and gradually learns to generate the whole sequence in one iteration. CTC learns to align the predictions with the targets extended by predefined rules.
> > We added the comparison with Imputer in the updated version (Section 4.2).
> >
> > &nbsp;
> >  &nbsp;
> >
> > * Jacob Devlin, Ming-Wei Chang, Kenton Lee, and Kristina Toutanova.  Bert: Pre-training of deep bidirectional transformers for language understanding. In NAACL-HLT, pp. 4171–4186, 2019.
> > * Alex Graves, Santiago Fernández, Faustino Gomez, and Jürgen Schmidhuber. Connectionist temporal classification: labelling unsegmented sequence data with recurrent neural networks. In ICML, pp. 369–376, 2006

---

### Official Review · AnonReviewer5 · 2020-11-02

**Rating:** 6
**Confidence:** 3

**Review:**

*Paper Contributions*

The paper proposes the Glancing Transformer (GLAT), a model for non-autoregressive generative model of language (focusing on the MT domain). GLAT incorporates several changes which result in a model which is state-of-the-art in several MT categories for non-autoregressive models and without requiring iterative sampling.

*Strong points of the paper*

* The paper clearly lays out the suggested model changes, and how they related to previous work.
* Some of the results are clearly state of the art for the task.
* The model and changes are described well and clearly.

*Weak points of the paper*

* GLAT incorporates several changes from the NAT-base architecture they primarily compare against, without clear ablations to attribute cause to the improvement.
* Most of the paper discusses the impact of architectures on inference/sampling speed, but doesn't discuss training speed, which seems relevant because my understanding of the model means that each step requires an additional decoder pass.
* Some of the writing is poor and overly-colloquial.

*Clearly state your recommendation (accept or reject) with one or two key reasons for this choice.*

I believe this paper is marginally above the acceptance threshold, though I think there is a clear avenue to improve the paper and change my rating to accept.

*Supporting arguments for your recommendation.*

The proposed GLAT model essentially consists of two changes from prior work:
1) Using an unconditional forward pass of the decoder to determine which tokens to mask, and then running the decoder a second time
2) Forwarding of encoding outputs via attention instead of mask tokens in the second decoder.

My main criticism of the paper is that is it unclear how each of the two changes contribute to the overall improvement being seen by the model. It might be the case that it is exclusively one of them, and I think the paper would clearly benefit from ablations discussing the two and showing how they each contribute.

As a secondary criticism, the paper proposes using a simple random sampling strategy to determine which tokens to mask (and which to glance at). While the authors defend this as the simple choice (and it certainly is), it feels to me that there is an obvious second alternative, which is to use the probability of the tokens generated by the decoder without glancing at anything (as in mask-predict's iterative sampling) to pick which tokens to glance at and which not to. It feels to me that this alternative is substantially obvious (seeing as it is exactly how mask tokens have been predicted in the past) that it should be attempted.

*Ask questions you would like answered by the authors to help you clarify your understanding of the paper and provide the additional evidence you need to be confident in your assessment.*

I would ask that two questions be answered, both ablations trying to get at the heart of the reason for improvement:

1) What happens if you do not forward the decoder inputs to the second decoding pass, but instead set them to a unique MASK token as in prior work (but keep all other aspects of training the same)?
2) What happens if you use mask-predict's probability-based sampling criterion to pick which tokens to sample, instead of the default random strategy.

Also, I have two questions which I think also should be answered, but don't require any further experimentation:

3) What is the impact on glancing sampling to training time?
4) What is the *exact* architecture of NAT-base? You show the encoder of GLAT works better than NAT-base - is NAT-base trained with the attention (instead of UniformCopy or SoftCopy)? Could that be the reason for the improvement?

I feel that if the authors answer these questions (ideally with WMT benchmark data for the first two), along with addressing the more minor points I list below, that I would feel comfortable increasing my rating to "accept".

*Additional feedback with the aim to improve the paper.*

I had a few minor questions while reading that I think would be useful to address:
* You mention your final model is based on averaging the 5 best checkpoints, how are you measuring best?
* You make a quite strong claim that "We think GLAT is superior to AT to some extent", which I think needs either more qualification or more focus.
* I struggled with understanding the paragraph starting with "Adaptive Sampling Number", I think it could use some clean-up.

Finally, I think most of the writing is clear, but there are a few points where it becomes quite colloquial and difficult to follow, I think an additional pass cleaning up some of the structure could be quite beneficial!

---

> ### Author Response · Authors · 2020-11-23
> **We have added ablation studies. (1)**
>
> Thanks very much for your insightful suggestions.
> As you mentioned in your questions, we add ablation studies to attribute causes to the improvements. Besides, we add discussions on training speed and answer all the minor questions.
>
> **Question 1: "What happens if you do not forward the decoder inputs to the second decoding pass, but instead set them to a unique MASK token as in prior work (but keep all other aspects of training the same)?"**
>
> We did a quick run and compare our proposed GLAT with Mask-Predict by:
> 1. We replace the glancing sampling with the uniform random sampling used by mask-predict
> 2. We use the MASK token as our decoder inputs but keep all other aspects of training the same
>
> |										|   |WMT14 EN-DE	|	WMT14 DE-EN |
> |--------------------------------------|---|:--------------------:|:---------------------:|
> |GLAT w/ sampling strategy of Mask-Predict	|	|	19.16		|		23.56	|
> |GLAT w/ decoder inputs of Mask-Predict 	|	|	24.99		|		29.48	|
> |GLAT									|	|	25.21		|		29.84	|
> |  |
>
> The results show:
> 1. the proposed glancing  sampling is crucial, without which the results will drop significantly (5 ~ 6 BLEU scores)
> 2. Replacing the decoder inputs with the MASK token will also lead to a small drop (0.2 ~ 0.3 BLEU). Although the performance drop is marginal, feeding representations from the encoder as the decoder input could still moderately improve the strong baseline after adopting the glancing sampling (23.56 ->29.48 -> 29.84 on DE-EN)
>
> To sum up, the adaptive glancing sampling approach contributes the most to the final improvement of GLAT, while the use of representations from the encoder also helps a bit.
>
> **Question 2: "What happens if you use mask-predict's probability-based sampling criterion to pick which tokens to sample, instead of the default random strategy."**
>
> Thanks for your kind notes.  We add experiments on WMT14 EN-DE, comparing mask-predict's probability-based sampling with the random sampling (the one used in our paper) to select reference words for glancing. Note that the two sampling strategies are used after the adaptive sampling number N is obtained by computing the Hamming distance.
> Here, "most certain" means we sample words at these positions with higher prediction confidence, and "most uncertain" means we choose those with the lower prediction confidence. Results are shown as follows:
>
> | 		|	|random	||	most certain	||	most uncertain	|
> |------------|---|:----------:|----------|:-------------:||:---------------:|
> |GLAT		||25.21	||	24.99		||	    24.86	|
> |GLAT(with reranking)||	26.55||		26.22	||			26.13|
> ||
>
> We find that all strategies give similar results and the random strategy performs a little better than the two probability-based strategies. We adopt the random strategy in GLAT because it is simple yet effective. Additionally, we think introducing randomness in the glancing sampling could enable GLAT to learn to explore dependencies among target words.
>
> **Question 3: "GLAT incorporates several changes from the NAT-base architecture they primarily compare against, without clear ablations to attribute cause to the improvement. "
> Question 4: "What is the exact architecture of NAT-base? You show the encoder of GLAT works better than NAT-base - is NAT-base trained with the attention (instead of UniformCopy or SoftCopy)? Could that be the reason for the improvement?"**
>
> The two questions are related, we answer them as follows:
>
> To Question3:
> Sorry for the confusion. The proposed GLAT and NAT-base only have a difference in training, they have the same model architecture. We have provided clear ablations in Question 1 to attribute causes to the final improvement.
>
> To Question4:
> 1. The model architecture of NAT-base is the same as Gu et al., (2018), except that a) we use attention to copy encoded representation as the decoder inputs instead of uniform copy. b) we do not use their fertility strategy;
> Note that NAT-base is also trained with attention copy. *Neither UniformCopy or SoftCopy is used in our implementation.*
> 2. Because NAT-base and GLAT are both trained with attention copy, the attention copy is not the reason that why the encoder of GLAT works better than NAT-base's. We think the improvements come from that after using the proposed glancing sampling strategy, parameters of GLAT is better trained than NAT-base. This leads to a better encoder in GLAT.
>
>  **Question 5: "Discussion on training speed"**
>
> Thanks for your kind suggestions. We added the discussion for training speed in our updated paper (Section 4.3). By training on the same device (8 V100 GPUs) for WMT14 EN-DE, NAT-base takes about 45 hours while GLAT takes about 56 hours. Compared to NAT-base, the training of GLAT is about 1.2 times slower, which is acceptable for achieving significant accuracy improvements.
>
> * Jiatao Gu, James Bradbury, Caiming Xiong, Victor O.K. Li, and Richard Socher. Non-autoregressive neural machine translation. In ICLR, 2018

---

> > ### Author Response · Authors · 2020-11-23
> > **We have added ablation studies. (2)**
> >
> > **Question 6: " Some of the writing is poor and overly-colloquial"**
> > Thanks, we have proofread the draft carefully and fixed these writing issues in the updated paper.
> >
> > **Question 7: "You mention your final model is based on averaging the 5 best checkpoints, how are you measuring best?"**
> > The best checkpoints are chosen by BLEU scores on the validation set. We made it clear in the updated version.
> >
> > **Question8: "You make a quite strong claim that "We think GLAT is superior to AT to some extent", which I think needs either more qualification or more focus."**
> > Sorry for the confusion, We have fixed it in the updated version.
> > Empirically, we find that GLAT outperforms AT on WMT14 DE-EN, when the length of the source input is less than 20. We speculate that it is because the modeling of target word dependencies is bidirectional in GLAT while AT is unidirectional (left-to-right). The bidirectional word dependency modeling introduces the great potential to parallel text generation models (similar to the differences between GPT and BERT for language understanding).
> >
> > **Question9: "I struggled with understanding the paragraph starting with "Adaptive Sampling Number", I think it could use some clean-up."**
> > Thank you for your advice, we revised this part in the updated paper to make it easy to follow (See Section 3.2).

---

### Author Response · Authors · 2020-11-23
**Paper Updates**

We thank all the reviewers for their valuable comments and suggestions.

We have individually responded to issues mentioned by reviewers and uploaded a revised draft with some major modifications as follows:
1. Fix grammatical errors and polish the paper for better delivery
2. Add experiments for ablation study (Section 4.4)
 > a. To explore more strategies for selecting reference words to replace the decoder inputs, we add four other reference word selection strategies in addition to the default random strategy for comparison
 > b. For exploring more metrics to compare the predictions and references, we add results of GLAT with Levenshtein distance along with the results of the Hamming distance
 > c. Add experiments for comparing GLAT and Mask-Predict:
    >>1. replace the decoder inputs of GLAT with [MASK] tokens
    >>2. replace the sampling strategy of GLAT with the uniform sampling strategy of Mask-Predict
3. Add discussions for  the training speed of GLAT (Section 4.3)
4. Provide more insights for the designing of GLM

We highly appreciate all reviewers for your efforts and time spent reviewing our paper and the constructive, critical, and meaningful suggestions for our method. Revisions are made in our submission to clarify our approach further and discuss these thoughtful suggestions.
We hope the above reply could address your concerns. Welcome to any further comments/questions!

---

### Decision · Program_Chairs · 2021-01-07
**Final Decision**

**Decision:**

Reject

**Comment:**

This work raised quite a few questions, and left the reviewers somewhat divided. The authors have done their best to answer these questions, conducting additional experiments where needed.

The close relation of this work to Mask-Predict (Ghazvininejad et al. 2019) was noted by several reviewers. Although the current version of the manuscript addresses this, the introduction still frames Mask-Predict as an iterative model, and does not explicitly make the connection between GLAT and single-iteration Mask-Predict. My impression is that this understates the relationship between these models somewhat.

Taking single-iteration Mask-Predict as a baseline, the proposed extension is fairly simple, and seemingly effective, which is a potentially impactful combination. However, the manuscript is still held back by presentation issues (including but not limited to spelling and choice of words), and I concur with Reviewer 2 that the connection with curriculum learning should be elucidated not just in words, but with supporting experimental analysis.

Regarding training cost, given that training for GLAT seems to be more costly for the same number of training iterations, a comparison where the total compute budget is held constant could be interesting -- though I appreciate that this is not a key point of the paper, as the authors point out (whereas inference cost is).

I believe the changes made by the authors in response to the reviewers' comments are substantial enough that they merit a further review cycle, and may still fall short of the reviewers' expectations in some aspects. Therefore, I will not recommend acceptance, though I want to add that this was a tough call to make. I would also like to encourage the authors to resubmit their updated manuscript.